# Ecological Restoration Practices within a Semi-arid Natural Gas Field Improve Insect Abundance and Diversity during Early and Late Growing Season

**DOI:** 10.3390/ani13010134

**Published:** 2022-12-29

**Authors:** Michael F. Curran, Joshua R. Sorenson, Zoe A. Craft, Taylor M. Crow, Timothy J. Robinson, Peter D. Stahl

**Affiliations:** 1Wyoming Reclamation and Restoration Center, University of Wyoming, Laramie, WY 82071, USA; 2Program in Ecology, University of Wyoming, Laramie, WY 82071, USA; 3Ecosystem Science & Management, University of Wyoming, Laramie, WY 82071, USA; 4Jonah Energy LLC, Pinedale, WY 82941, USA; 5Department of Plant Science, University of California–Davis, Davis, CA 95616, USA; 6Department of Mathematics and Statistics, University of Wyoming, Laramie, WY 82071, USA

**Keywords:** arthropods, ecosystem services, entomology, land reclamation, pollinator, richness, Rocky Mountain bee plant, yarrow

## Abstract

**Simple Summary:**

Land reclamation and ecological restoration are required to mitigate land surface disturbances associated with natural gas extraction in the western United States. Traditional focus on these lands has been to stabilize soil to prevent erosion, though more recently, there has been an emphasis on restoring ecosystem services. Insects provide numerous ecosystem services and can be considered indicators of success for ecological restoration projects. It has been suggested that creating spatial and temporal mosaics of flowering plants will be necessary for pollinator conservation. In this study, we found insect abundance to be significantly higher on sites undergoing restoration than in adjacent reference areas during early and late growing season within the Jonah Infill natural gas field in Sublette County, Wyoming, USA. Family richness was significantly higher on well pads in the late season compared to reference sites. While our study only looked at sites within the Jonah Infill and had limited scope in terms of seed mixes, our results are promising and suggest that further work be done to investigate the efficacy of flowering plants on increasing insect abundance and richness in future restoration projects associated with oil and natural gas development.

**Abstract:**

Insects are critical components of terrestrial ecosystems and are often considered ecosystem engineers. Due to the vast amount of ecosystem services they provide, because statistically valid samples can be captured in short durations, and because they respond rapidly to environmental change, insects have been used as indicators of restoration success and ecosystem functionality. In Wyoming (USA), ecological restoration required on thousands of acres of land surface have been disturbed to extract natural gas. In this study, we compared early seral reclamation sites to reference areas at two points within a growing season. We compared insect abundance and family richness on 6 natural gas well pads with early season perennial forbs and 6 well pads with the late season to insect communities on adjacent reference areas. A total of 237 individual insects were found on early season reclaimed sites compared to 84 on reference sites, while 858 insects were found on late season reclaimed sites compared to 38 on reference sites. Insect abundance was significantly higher on reclaimed well pads compared to reference areas at both points in the growing season, while reclaimed sites had significantly higher Shannon Diversity Index in early season and significantly higher family richness in late season compared to their paired reference sites. We also found interesting differences in abundance at family levels.

## 1. Introduction

Insects, the most abundant and diverse group of animals in the world, provide numerous ecosystem services [1,2]. Insects are often considered ecosystem engineers because they impact soil properties [3], influence nutrient cycling [4], serve as a food source to higher trophic levels [5], and provide critical pollination services allowing plants to successfully reproduce [6,7,8]. Since they provide more ecosystem services than other wildlife, respond rapidly to environmental change, and because statistically valid samples can be captured in a short duration, insects have been used as indicators of restoration success [9,10,11]. A meta-analysis conducted in 2011 suggests the primary focus of insects in ecological restoration projects had been towards pollinators in agriculture systems [12]. Since then, studies related to insect restoration have become more common in non-crop ecosystems, perhaps because of the ecosystem services insects provide [2]. Habitat restoration efforts have shown a general trend towards benefiting pollinating insects [13] and examples where utilizing native plants in restoration efforts have improved insect habitat include urban/suburban ecosystems [5], mine sites [8], wildfire locations [11], and natural gas well pads [14]. 

In the western United States, extraction activities related to energy development have resulted in hundreds of thousands of acres of land surface disturbance. In Wyoming, oil and gas operating companies are required to reclaim surface disturbances after extraction activities take place while complying with multiple regulatory criteria [15,16]. These criteria have typically focused on establishing vegetation deemed suitable by regulatory agencies with a heavy emphasis on controlling erosion [17]. All lands undergoing reclamation associated with oil and gas disturbance in Wyoming require operators to establish 70% or greater relative ground cover compared to an adjacent reference area to comply with the Wyoming Department of Environmental Quality’s Stormwater Pollution Prevention Plan [16], though this State Government requirement typically does not have species-specific vegetation guidelines. On private lands, operating companies are instructed to work with landowners to develop specific reclamation plans, whereas on Federally owned Bureau of Land Management (BLM) land, operators are required to comply with various BLM Field Office regulatory criteria [15,16,17]. These criteria often contain metrics to ensure reclamation efforts minimize or eliminate vegetation considered noxious and invasive weeds, stabilize site locations for erosion control, and several BLM Field Offices require reclaimed locations to meet or exceed native vegetation species richness compared to adjacent reference sites [16]. More recently, focus has shifted from strictly land reclamation towards restoring disturbed landscapes to functional ecosystems, providing suitable wildlife habitat [18]. Insects and other arthropods, as wildlife which serve as ecosystem engineers and play significant roles as bottom-up drivers of trophic interactions [19], have been used as indicators to determine success of ecosystem restoration projects [10], but have rarely been studied in oil and natural gas fields [14,20]. 

The first study to examine how insects respond to ecosystem restoration efforts associated with natural gas development in a sagebrush-steppe ecosystem suggests these efforts increase insect abundance and richness when comparing well pad locations which were seeded with either native grasses or forbs to adjacent reference communities [14]. The mass flowering hypothesis, which suggests dense patches of flowering plants are likely to increase pollinator abundance in areas where the surrounding landscape is void of diverse floral vegetation [21,22,23], was tested in that study. A limitation of the study, which occurred in the Pinedale Anticline natural gas field, is it only examined one flowering species, Rocky Mountain bee plant (*Cleome serrulata*; hereafter RMBP) at one point in the growing season. This issue is found among studies which have tested the mass flowering hypothesis in agricultural systems. Research suggests having a diverse spatial and temporal mosaic of flowering plants which bloom at different periods throughout the growing season is an effective strategy for insect and pollinator conservation e.g., [24]. 

In the Jonah Infill natural gas field (hereafter Jonah Field), well pads are constructed by stripping topsoil and vegetation and stockpiling it to allow for drilling equipment to operate on level surface. Typically, 70–80% of initial land surface disturbance has soil respread and seeded the same year construction equipment is removed from the site. Sites in this study were seeded with the annual forb, RMBP, and a mix of other native perennial forbs, shrubs, and grasses. It is common for the annual forb, Rocky Mountain bee plant, which grows well in disturbed soil [25] to be the most prolific species in terms of establishment and flowering in the first year after reclamation [26]. However, it tends to become less dominant within two-three years as perennial forbs, shrubs and grasses establish [26]. While RMBP flowers later in the season, many perennial forbs in the seed mix flower earlier in the season. 

The objectives of this study, which took place in the Jonah Field (Sublette County, WY, USA), were: (1) to compare insect communities on well pads undergoing reclamation and restoration activities to adjacent reference communities, and (2) to determine if differences in insect communities existed between early and late season well pads and reference sites.

## 2. Materials and Methods

### 2.1. Study Area

A total of 12 well pads undergoing interim reclamation and their adjacent reference areas were monitored in the Jonah Infill natural gas field in Sublette County, WY, USA. This field is predominantly Federal land and is regulated by the Jonah Interagency Office (JIO–consisting of Pinedale BLM Field Office, Wyoming Department of Agriculture, WYDEQ, and Wyoming Game and Fish Department). Well pads in this study ranged from 4.64–5.57 acres (mean = 5.33 acres) of initial surface disturbance with 80% (mean = 4.26 acres) of that area undergoing interim reclamation (i.e., soil respread and seeded on non-active portions of the well pad). 

All well pads were located in the Stud Horse Butte (SHB) section of this gas field. All sites were located in ecological site R034AC122WY. This ecological site description is dominated by loamy soils. The historic climax plant community is described as ‘big sagebrush/bunchgrass’, though the state and transition model for the site suggests continuous high intensity early season grazing may change the historic climax plant community to ‘big sagebrush/bareground’. Grazing records were unavailable, though it has been documented the area has been grazed primarily in the Spring since the early 1870s [27]. This site description averages between 30 and 70 frost free days per year, with mean annual temperatures between 1.1–3.3 °C. The average annual precipitation, according to the ecological site description, is 22.9 cm–30.5 cm. The elevation of the area ranges from 2136 m–2219 m. 

All well pads were seeded with seed mix ‘B1’ (Appendix A). The mix contained four native shrub species, 10 native perennial forb species, one annual forb species, and six native perennial grass species. All seeds were regionally sourced and purchased from a local seed vendor. Additionally, all species are native to Sublette County, WY. This seed mix was approved by the BLM in accordance with the 2006 Jonah Infill Drilling Project Record of Decision to meet desired native vegetation communities capable of supporting the multiple use mandate of the agency to account for wildlife and domesticated livestock habitat. Six of the well pads were 3 years old and consisted mainly of native perennial forbs, grasses and shrubs. Six of these sites were sampled on 22 June 2017 while perennial forb species were in bloom. The other six sites in this study were in their first year of reclamation and were sampled on 27 July 2017 when the native annual forb, RMBP, was in bloom as the dominant vegetation species.

### 2.2. Vegetation Sampling

Well pads which consisted of early season blooming flowers were sampled on 22 June 2017, while later season flowers were sampled on 27 July 2017. Our study design was a matched pair design, with two forty meter transects sampled at 5 m and 10 m away from the edge of the well pad, both in the reference area and on the well pad (Appendix A). Reference site selection was in accordance with the 2006 Jonah Infill Drilling Project Record of Decision. A 0.5m^2^ image was taken at the start of each transect and at 5 m increments along the transect, resulting in 9 images per transect, or 18 for each well pad and associated reference area. The images were taken using a free-hand nadir technique [28] and had ground sample distances of ~0.4mm which is similar to previous studies examining vegetation on reclaimed well pads and rangelands [16,29,30]. A 7×7 grid was placed in each image in ‘SamplePoint’ [31]. This resulted in a total of 49 pixels being classified per image, or 441 pixels per transect (882 pixels per each well pad and associated reference area). Pixels were classified to the species-specific level for shrubs and forbs, while grasses were classified as rhizomatous or bunch grass. Bareground, rock, and litter were also classified in pixel analysis.

### 2.3. Insect Sampling

Insect sampling was conducted on the same days as vegetation sampling with a matched pair design. Similar to vegetation sampling, insect sampling was conducted when flowering plants on the selected well pads were in bloom. Two runs of forty sweeps were taken along the vegetation transect locations on reclaimed well pads and adjacent reference areas. A sweep net with a 38 cm diameter and a 50 cm handle was used to collect insects. All sampling was conducted between the hours of 730 and 1600 in sunny conditions with winds < 8 kph. In all instances, two individuals walked along the transect, with the lead individual conducting the sweep net collection and the follower taking freehand nadir images as described above. After each transect, technicians walked at least 250 m away from the sampling area to transfer insects from the sweep net into a Zip-Lock^®^ bag and then into a cooler equipped with dry ice for preservation where a minimum of a 5-minute wait period was conducted by both individuals before walking back to the next transect location to avoid ‘flushing’ effects on insects [32]. 

Upon return to the University of Wyoming (Laramie, WY, USA) from the field, samples were placed in a freezer for laboratory identification at a later date. As only adult insects were collected in our study, they were identified to family level under a microscope with aid from National Audubon Society field guide [33]. Family-level diversity is typically similar to species-level diversity in terrestrial ecosystems [11,34,35]. 

### 2.4. Statistical Analysis

Spreadsheet reports generated from SamplePoint analysis were used to classify vegetation cover into bare ground, litter, and vegetation by species. Vegetation was classified into the following functional groups: forb, grass, or shrub. Wilcoxon signed-rank tests on paired differences were used to compare percent cover for bare ground, litter, forb, grass, and shrub between reclaimed sites and reference sites for each sampling period. Since these sampling events took place at different time points, early season sites and late season sites were compared separately for both insect and vegetation analyses. 

For insect community analysis, sites were separated into the following groups before comparison: (1) 3-year-old well pads monitored in early season, (2) reference sites adjacent to early season sites, (3) 1-year-old well pads monitored later in season, and (4) reference sites adjacent to later season sites. A Poisson random effects model (log-link) was used to make comparisons between insects on reference vs. reclaimed sites in early and late season as well as comparisons between insects within reclaimed and reference sites in early and late season. Shannon Diversity Index values were created for each location and averaged on a per-group basis. Wilcoxon tests were used as a convenient summary to compare individual families within each of the original 4 contrasts.

## 3. Results

### 3.1. Vegetation Sampling

Vegetation communities on transect locations on reclaimed well pads differed from reference areas (Figure 1). Percent forb coverage was significantly greater on reclaimed well pads than reference areas (*p* < 0.001) where no forbs existed in either the early or late season collection period (Figure 1). While 100% of forb coverage on reclaimed sites in the later growing season was RMBP, upwards of 95% of forb coverage on early season sites was western yarrow (*Achillea millefolium*). Other forbs on early reclaimed sites included Lewis blue flax (*Linum lewisii*), Palmer’s penstemon (*Penstemon palmeri*), littleflower penstemon (*Penstemon procerus*), scarlet globemallow (*Sphaeralcea munroana*), and evening primrose (*Oenothera pallida*). Percent shrub coverage was significantly greater on reference areas than reclaimed sites (*p* < 0.001) where shrubs were not present within transect locations (Figure 1). All shrubs in the reference areas were Wyoming big sagebrush (*Artemisia tridentata spp. wyomingensis*). Percent litter coverage was significantly higher on reclaimed sites than reference areas in the early season (*p* = 0.0355). No other significant differences existed between percent coverage categories (Figure 1). Although it appears as percent grass cover was significantly greater on early season reclaimed sites than reference areas, a *p*-value = 0.0591 suggests the sites were not significantly different under *p* = 0.05 criteria.

### 3.2. Insect Sampling

Insect abundance was significantly higher (*p* < 0.0001) among site types with more individual insects found on reclaimed areas than adjacent reference areas (Figure 2). The mean number of individual insects found on early season well pads was 39.5 (standard deviation = 14.58) compared to 14 (standard deviation =7.38) on adjacent reference sites, meanwhile the mean number of insects on late season well pads was 143 (standard deviation = 60.66) compared to 6.33 (standard deviation=3.2) on their reference sites. A total of 237 individual insects were found on early season reclaimed sites with 83 on the reference sites, whereas 858 insects were found on late season sites and 38 were found on the reference sites (Appendix A).

While mean insect family richness was higher on early season reclaimed sites (8.33) than reference sites (5.17), the difference was not statistically significant. However, mean Shannon Diversity Index values were significantly higher (*p* = 0.038) on early season reclaimed sites (H = 1.74) than reference sites (H = 1.42). In the late season, mean insect family richness on reclaimed sites (9.5) was significantly higher than reference sites (3.83, *p* = 0.036). These findings are corroborated by family richness accumulation curves which show overlapping error bars when comparing early season reclaimed sites to early season references sites, but no overlapping error bars when comparing late season reclaimed sites to late season reference sites (Figure 3). 

A total of 24 insect families were identified in this study. A total of 18 insect families were found in the early season collection and 15 were found in the late season collection (Figure 4). Of these, nine families were common to both collections (Braconidae, Chalcididae, Cercopidae, Coccinellidae, Chrysomelidae, Formicidae, Halictidae, Lygaeidae, and Sarcophagidae). Insect abundance within families were not significantly greater in reference areas than reclaimed areas during either early or late season collections (Figure 3). In early season, Phoridae and Miridae were significantly more abundant on reclaimed sites than reference areas (*p* < 0.05; Figure 4). In late season, Chrysomelidae, Lygaeidae, Bombyliidae, Halictidae, and Formicidae were significantly more abundant on reclaimed well pads compared to reference areas (*p* < 0.05; Figure 4). 

## 4. Discussion

As predicted, we found more insect abundance and richness on restoration sites than the surrounding reference system. While we cannot rule out climatic factors as a potential cause for there being fewer insects in the first collection, previous research has shown that yarrow, penstemon and blue flax, which were the main flowering species in bloom during the first insect collection, attract less wild bees than other native perennial forbs which bloom around the same time [36]. However, it is not surprising that the well pads with early blooming flowers contained greater insect abundance and richness than the adjacent reference system which had zero flowering species and were predominately composed of decadent sagebrush and bareground. Our results from late season sampling corroborate the previous study to examine insects on well pads containing Rocky Mountain bee plant and adjacent reference sites [14]. 

While we were not able to determine if insects were moving among well pads at either sampling time of our study, it is possible that providing beneficial resources (i.e., flowers) in a patchy matrix throughout the existing landscape could improve insect movement, reproduction, and increase overall abundance and richness of insects in the area [37]. If insects are, in fact, moving among well pads, they may be delivering benefits to plants in the reference areas by increasing genetic diversity through cross-pollination and may be providing other ecosystem services such as comminution of organic matter, nutrient cycling and acting as a food source for other birds and animals [19]. Although vegetation communities were different between reclaimed sites and reference areas, with reference areas exhibiting vegetation communities closer to climax stages of succession, it is important to understand ecosystem dynamics on early seral reclamation sites because land restoration in arid environments is challenging [38] and sagebrush reestablishment on disturbed sites may take decades [39].

While there was significantly more RMBP cover on late season well pads than there was overall flower cover on early season well pads, no studies have set a minimum threshold for minimum percent cover of a flowering species to constitute ‘mass flowering.’ As there were no forbs found in reference sites in the early season vegetation monitoring, we assume the restoration areas containing flowers are positively drawing in insects based on research suggesting flower color, scent and height all play roles in being insect attractants [e.g., 32]. This is the first study to compare insects at more than one point in a growing season on well pads undergoing restoration activities while flowering events are occurring with distinctly different vegetation species. While a total of nine out of 24 insect families found across the two sampling periods were common to both collections, we were unable to determine if insects from early sites traveled to later sites or if unique insects at each time period were found due to specific life history characteristics of those families. As insect dispersal studies are becoming more common and affordable [40], future research into this area may be beneficial to overall conservation and restoration efforts for insects in oil and natural gas fields. Additionally, determining if insects, which are attracted to these restoration efforts, are acting as food sources to vertebrate species or providing other ecosystem services to the reference community would be beneficial. This could potentially result in well pads acting as positive reserves instead of negative fragments, making them critical to insect conservation [37]. Finally, increasing the amount of native forb seeds which can be bought commercially may help improve restoration projects, both by having new plant species increasing ground cover on disturbed sites, and by helping conserve or restore insect habitat and populations and allowing them to continue to provide critical ecosystem services [36,41,42].

Although this study was limited to one year, the results were highly significant and will help direct future research related to insect response to reclamation and restoration activities related to oil and gas development. Additionally, the study is limited in that insects were identified to family, rather than species-level. While insect richness at the family-level is closely linked to insect richness and the species-level [34,35], future studies which identify insects to species will benefit the overall understanding of insect communities in the area. Finally, this study was limited to one method of insect sampling (sweep netting), which is biased towards capturing flying insects or other insects within the canopy of the vegetation. It is to be expected other insect sampling techniques would yield different results [43]. Future studies using pitfall traps would be beneficial to understand how ground-dwelling insects respond to reclamation and restoration within the Jonah Field. Studies to determine how insects are moving among vegetation patches in natural gas fields would also be beneficial in determining the role restoration plays in providing benefits to reference areas.

## 5. Conclusions

Natural resource extraction is expected to continue into the foreseeable future with growing populations. Reclamation and restoration of areas disturbed for energy extraction will become increasingly critical to restore habitat for wildlife species including insects, and therefore is important to overall insect conservation. This study suggests that adding native, flowering plants to reclamation efforts within a sagebrush-steppe ecosystem results in increases in insect abundance and family richness compared to reference areas when the reference areas are devoid of flowering plants. Continued use of native forb species in reclamation and restoration efforts will be important for insect and pollinator conservation in the western United States and other areas where anthropogenic activities result in land surface disturbance.

## Figures and Tables

**Figure 1 animals-13-00134-f001:**
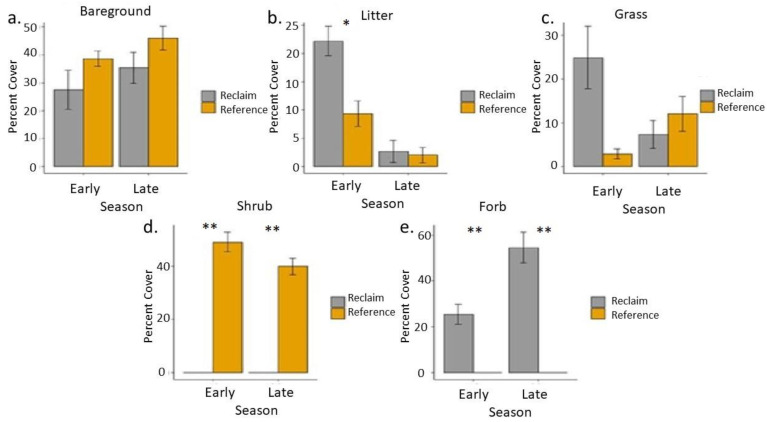
Bar graphs showing percent cover of (**a**) bare ground, (**b**) litter, (**c**) grass, (**d**) shrubs, and (**e**) forbs on reclaimed and reference sites which were monitored in early season and late season. * *p*-value < 0.05, ** *p*-value < 0.001.

**Figure 2 animals-13-00134-f002:**
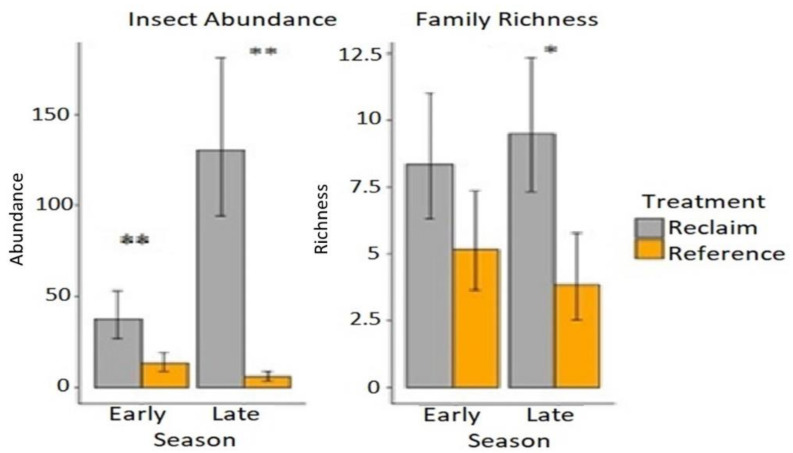
A graphical representation of species abundance (left) and species richness (right) comparing reference vs. reclaimed within early season and within late season (abundance; *p*-value <0.0001 ** & richness; *p*-value < 0.05 * (Late); *p* =0.2019 (Early).

**Figure 3 animals-13-00134-f003:**
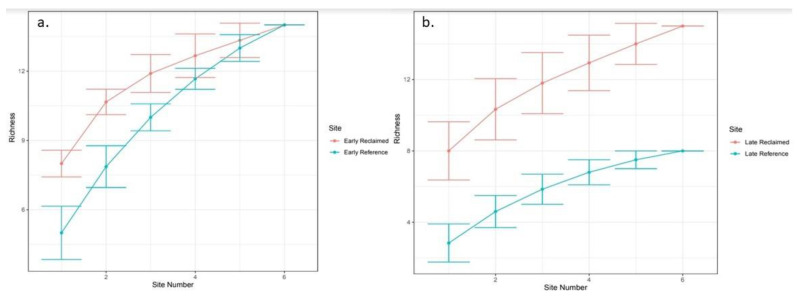
Accumulation curves with Family Richness on the *y*-axis and Site Number on the x-axis for Early Season (**a**) and Late Season (**b**).

**Figure 4 animals-13-00134-f004:**
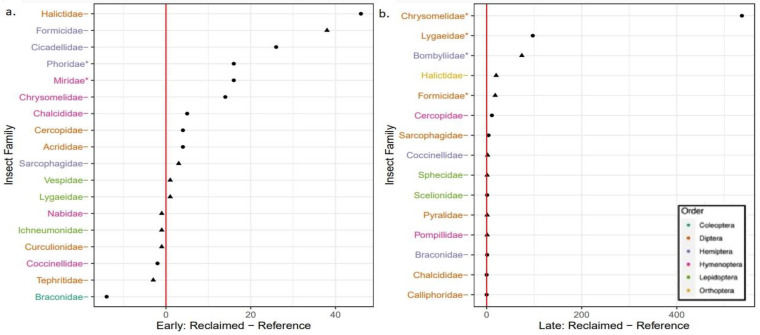
Total family level difference within each of 4 contrasts. (**a**) Reclaimed minus Reference within Early season. (**b**) Reclaimed minus Reference within late season. Triangular points represent families that were only found in one of the two conditions, whereas circular points represent families found in both conditions. Families with * next to them are those which had significant differences between site type with *p* < 0.05.

## Data Availability

The data presented in this study are available in the manuscript and in Appendix A.

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
