# Peer review of "Ecological Restoration Practices within a Semi-arid Natural Gas Field Improve Insect Abundance and Diversity during Early and Late Growing Season"

_animals, 2022, doi:10.3390/ani13010134_

Round 1

Reviewer 1 Report (Previous Reviewer 2)

The authors have integrated most of my minor suggestions but not the major ones.

1.       Even if other studies were based on family-level diversity, this does not mean that this is the correct approach for tackling the research question in your study. From your comment, I surmise that you do not want/cannot enhance the taxonomic resolution of your study. Therefore, my suggestion is that throughout the text (including the title), you change “diversity” to “insect family diversity” and recognise this limitation (the taxonomic resolution) in the discussion.

2.       Statistically significant results can be spurious when the sample size is small and are not an indication of quality. The authors claim this is a community-level study, but they sampled only for two days. Then they should create accumulation curves (at morphospecies level at least) to convince the reader that the insect community sampled was indeed representative of the actual community.

3.       The comment regarding the Study Area was not tackled. Please add a map of the study area marking your sampling points. If the sampling points are between 5 and 10 m between the two sites, then there is no way that they can be considered independent.

4.       OK, but then we have two sites sampling at different times. If there are differences, are they due to the different sampling periods or to the different locations?

5.       I am sorry, but the argument that this is a short communication does not justify insufficient sampling effort. In order to understand the real effects of this flower mixture and discuss the results in the most appropriate way, additional field seasons are necessary.

6.       If the sampling points were just 5-10 m from each other, then they cannot be independent, and no statistical tool will fix that.

7.       There was also an unaddressed comment (previous L229-234). How can you compare early and late seasons if the sampling sites were not the same?

8.       Regarding the raw data, you could combine the two tables (one on top of the other) by adding “Early season” and “Late season”.

Author Response

Reviewer 1

The authors have integrated most of my minor suggestions but not the major ones.

  1. Even if other studies were based on family-level diversity, this does not mean that this is the correct approach for tackling the research question in your study. From your comment, I surmise that you do not want/cannot enhance the taxonomic resolution of your study. Therefore, my suggestion is that throughout the text (including the title), you change “diversity” to “insect family diversity” and recognise this limitation (the taxonomic resolution) in the discussion.

We added ‘family richness’ to the title.  The word ‘family’ was already placed in front of diversity or richness in all instances throughout the manuscript based on comments from previous reviewer in the last round, therefore we did not make this change again.

We already stated in the discussion: “Additionally, the study is limited in that insects were identified to family, rather than species-level.  While insect richness at the family-level is closely linked to insect richness and the species-level [34-35], future studies which identify insects to species will benefit the overall understanding of insect communities in the area.”  Therefore, we did not change that again.

  1. Statistically significant results can be spurious when the sample size is small and are not an indication of quality. The authors claim this is a community-level study, but they sampled only for two days. Then they should create accumulation curves (at morphospecies level at least) to convince the reader that the insect community sampled was indeed representative of the actual community.

Spurious effects are certainly a concern in studies where data dredging and complicated models are fit. Here, we only compare the reference to the reclaimed sites and we do this separately for early season and late season. The models are simple hierarchical models that allow us to account for the spatial correlation of reclaimed to reference locations. Given the fact that our model is simple and that the hypotheses are scientifically defensible, there is no reason to believe that the p-values are spurious. 

The only time we mentioned insect community levels was line 192.  We have changed “For insect community analysis….” To “For insect abundance and family richness analysis…”

We have created accumulation curves (at the family level) based on your comment and R3’s comments. 

  1. The comment regarding the Study Area was not tackled. Please add a map of the study area marking your sampling points. If the sampling points are between 5 and 10 m between the two sites, then there is no way that they can be considered independent.

We have added a map of the area as supplemental file along with some photos to help visualize.  We sampled insects within reclaimed well pad boundaries and in adjacent, undisturbed reference areas (as noted in the manuscript, in accordance to local regulatory policy).  While our sweep net samples and vegetation transects were in close proximity, we addressed independence by putting a wait period between sampling to avoid flushing effects, which is similar to methods of Wenninger & Inouye (which we cite) which has been cited 139 times.  We also address this in our response to your 2nd comment.

  1. OK, but then we have two sites sampling at different times. If there are differences, are they due to the different sampling periods or to the different locations?

Thank you. This point is well-taken.  We have removed Panels C & D from Figure 3 (now Figure 4 due to the accumulation curve figure) as well as the discussion about differences between the sites at different times.

  1. I am sorry, but the argument that this is a short communication does not justify insufficient sampling effort. In order to understand the real effects of this flower mixture and discuss the results in the most appropriate way, additional field seasons are necessary.

The author guidelines suggest: “Communications of preliminary, but significant, results will be considered.”  Our study found highly significant differences in insect abundance and interesting differences among insect families.  This is the first study to examine insects in this area and the 2nd study to address insect response to restoration efforts of natural gas well pads in general.  The significant findings in the results, though preliminary (based on one year of data), warrant publication as a Communications article (rather than a Research Article) as they suggest companies required to restore their disturbances may have positive impacts on insects if they include native flowering species in their seed mixes.  The results of this study, clearly, can lead and help guide future, more long-term research.  Additionally, the findings from the late season section of this study corroborate a multi-year study (also from Sublette County, WY but in a different gas field) which was published in Land earlier in 2022.  This is stated at the end of paragraph 1 in the Discussion.

  1. If the sampling points were just 5-10 m from each other, then they cannot be independent, and no statistical tool will fix that.

While our sweep net samples and vegetation transects were in close proximity, we addressed independence by putting a wait period between sampling to avoid flushing effects, which is similar to methods of Wenninger & Inouye (which we cite) which has been cited 139 times. Additionally, we address your concern about independent samples in response to your 2nd comment.

  1. There was also an unaddressed comment (previous L229-234). How can you compare early and late seasons if the sampling sites were not the same?

We have removed this comparison (as noted in our response to your 4th comment).  Thank you for this keen observation.

  1. Regarding the raw data, you could combine the two tables (one on top of the other) by adding “Early season” and “Late season”.

We have added 1 .xlsx file with multiple tabs to address this

Reviewer 2 Report (Previous Reviewer 3)

Thank you for submitting your revised manuscript. I am satisfied that my earlier comments have been addressed and look forward to seeing the final published version of this article.

Author Response

Thank you!

Reviewer 3 Report (New Reviewer)

This study has multiple shortcomings (one year, family-level identification, sampling method), and the authors admit them, which I appreciate. Although I think these shortcomings may be relatively easily overcome (by adding another sampling year, identification to the species level, or at least to morphotypes, …) to obtain more reliable and generalizable results, I understand that this study should be perceived as a preliminary insight into the impacts of restoration of gas fields on insect communities. However, having all these shortcomings in mind, conclusions of this study are overstated. Moreover, as it currently stands, the manuscript is hard to follow in certain parts and needs to be improved. I am also concerned about statistical approach in certain points. Specific comments are listed below.

Major comments:

224 Reporting and interpreting insect abundance is misleading – it could be very few individuals in some families. This is evident also in lines 246–7. I think calculating Shannon’s index (although family-based) would be more informative here.

234 – species richness: is it comparable based on your sampling design? Species accumulation curves of family richness might be more appropriate here

Minor comments:

L 54-55  “because statistically valid samples can be captured in a short duration” is overstated. Short-term monitoring may often yield highly distorted results (and that’s why I said that your conclusions are overstated).

L 137–138 What was the link between the vegetation seeds and the composition revealed in your study?

L 166 What was the weather like during the smpling? Sweeping – what groups were you aiming at?

L208 Linus or linum palmeri? (discrepancy in supplementary table and manuscript)

L212–213 Italics

224 – add test statistics, p-value is not enough, the same applies to L249, 251, 253–5

L 232 – I was not able to locate the Supplemental Documents 2 & 3

Fig. 2 – levels of significance: indicate them in the figure (p-value or asterisk)

239-240 – statistically not significant = it was not higher

243 – the link to the fig. 2 is not located correctly

L246–256 “Insect abundance within families were not significantly greater in reference areas than reclaimed areas during either early or late season collections (Figure 3).” I don’t understand – after this statement, you name several taxa whose abundance was significantly higher in reclaimed areas either in early or late season. Also, why are certain families more/less abundant in early/late season in reclaimed/reference sites? There isn’t any further comment on these patterns in the discussion. Are they related to their life history? I think that a discussion on this topic is necessary.

Colors in fig. 3 are very hard to follow – add more contrast

L251 – I have no access to supplemental figure 2

L253 – “while the opposite was true for Phoridae…” what is the opposite? This is very hard to follow.

264 – but only at late season reclaimed sites, right? You cannot claim the richness was higher at early reclaimed sites if it was not significant (the same for line 270)

265-266 – exactly; that’s why it is quite dangerous to make conclusions from one sampling point

275-287 but you don’t know if they were moving among patches, right? Given that you killed your specimens. So maybe there could be some discussion on the methods and importance of studying movement among patches?

L 296-297 Could you expand on dispersal abilities of the shared insect families?

L 296 – “found across the two studies” – please revise

Figure 2 – b) says species richness, but L 242-243 is about family richness, right? So why do you present species richness in the figure?

326 and onwards – this study convincingly shows – too strong, given your sampling design, your study is only a preliminary insight showing conservation potential of reclamation sites

Author Response

Reviewer 3

This study has multiple shortcomings (one year, family-level identification, sampling method), and the authors admit them, which I appreciate. Although I think these shortcomings may be relatively easily overcome (by adding another sampling year, identification to the species level, or at least to morphotypes, …) to obtain more reliable and generalizable results, I understand that this study should be perceived as a preliminary insight into the impacts of restoration of gas fields on insect communities. However, having all these shortcomings in mind, conclusions of this study are overstated. Moreover, as it currently stands, the manuscript is hard to follow in certain parts and needs to be improved. I am also concerned about statistical approach in certain points. Specific comments are listed below.

Thank you for your comments.  We have responded below and made numerous changes to improve this manuscript based on your comments.  We believe we have addressed and admitted our study limitations and, as noted to Reviewer 1, have submitted this as a communications paper to report important findings rather than as a full research paper.

Major comments:

224 Reporting and interpreting insect abundance is misleading – it could be very few individuals in some families. This is evident also in lines 246–7. I think calculating Shannon’s index (although family-based) would be more informative here.

We included Shannon Index in our first draft, though we removed it based on reviewer comments before submitting our second draft.  We found significantly higher Shannon Index on early season reclamation sites compared to their reference, but no significance in the late season.  We appreciate your comment, but believe reporting abundance and richness (the simplest measure of diversity) is sufficient as the goal of our study was to determine if there was differences in insect abundance and family richness between reclaimed and reference sites.

234 – species richness: is it comparable based on your sampling design? Species accumulation curves of family richness might be more appropriate here

We have added family richness accumulation curves (in addition to reporting richness).  We believe richness is comparable on our sampling design, as we sampled the reference areas the same as we sampled our reclaimed areas.

Minor comments:

L 54-55  “because statistically valid samples can be captured in a short duration” is overstated. Short-term monitoring may often yield highly distorted results (and that’s why I said that your conclusions are overstated).

This is a reason why Longcore 2003 (cited 274 times) gives for insects being valuable to study in restoration settings.  We cite Longcore at the end of this sentence.  We appreciate your comment and hopefully (based on responses to your comments below) have addressed this in our discussion.

L 137–138 What was the link between the vegetation seeds and the composition revealed in your study?

We discuss this in the results lines 203-219.

L 166 What was the weather like during the smpling? Sweeping – what groups were you aiming at?

We have added ‘in sunny conditions with winds <8 kph’ at line 172.

L208 Linus or linum palmeri? (discrepancy in supplementary table and manuscript)

Linum, thank you for the keen observation!!

L212–213 Italics

Fixed, thanks!!

224 – add test statistics, p-value is not enough, the same applies to L249, 251, 253–5

These are included in the following sentences or sentence leading up to p-value being reported.

L 232 – I was not able to locate the Supplemental Documents 2 & 3

Thank you, we have addressed this (there are 2 supplemental documents, we mislabeled this in the manuscript on our revision…. And it appears as though we forgot to upload a document in the revision which was included in the first draft).

Fig. 2 – levels of significance: indicate them in the figure (p-value or asterisk)

Fixed, thanks!

239-240 – statistically not significant = it was not higher

We reported the mean to address this.

243 – the link to the fig. 2 is not located correctly

Fixed, thanks!

L246–256 “Insect abundance within families were not significantly greater in reference areas than reclaimed areas during either early or late season collections (Figure 3).” I don’t understand – after this statement, you name several taxa whose abundance was significantly higher in reclaimed areas either in early or late season. Also, why are certain families more/less abundant in early/late season in reclaimed/reference sites? There isn’t any further comment on these patterns in the discussion. Are they related to their life history? I think that a discussion on this topic is necessary.

Because no insect families had abundance higher in reference areas than reclaimed sites… but several families in reclaimed areas were significantly higher than reference areas.

We removed our statistical comparisons across seasons, but added a line in the discussion suggesting differences in families may be due to life history characteristics of these insect families.

Colors in fig. 3 are very hard to follow – add more contrast

We removed panels C & D and believe the smaller figure is easier to read.

L251 – I have no access to supplemental figure 2

We have added the correct supplemental files.

L253 – “while the opposite was true for Phoridae…” what is the opposite? This is very hard to follow.

We have removed this from the manuscript and agree with you and R1 that the last 2 sentences of that paragraph were hard to follow due to the odd comparison between early and late season.  Thank you.

264 – but only at late season reclaimed sites, right? You cannot claim the richness was higher at early reclaimed sites if it was not significant (the same for line 270)

We have removed this (see above and comments to R1)

265-266 – exactly; that’s why it is quite dangerous to make conclusions from one sampling point

We have removed this (see above and comments to R1)

275-287 but you don’t know if they were moving among patches, right? Given that you killed your specimens. So maybe there could be some discussion on the methods and importance of studying movement among patches?

Correct, which we stated at the beginning of the paragraph.  We have added a sentence at the end of the discussion to suggest future research studying insect movement will be beneficial.

L 296-297 Could you expand on dispersal abilities of the shared insect families?

L 296 – “found across the two studies” – please revise

We changed ‘studies’ to ‘sampling periods.’  Thank you.

Figure 2 – b) says species richness, but L 242-243 is about family richness, right? So why do you present species richness in the figure?

Fixed, thank you!

326 and onwards – this study convincingly shows – too strong, given your sampling design, your study is only a preliminary insight showing conservation potential of reclamation sites

Thank you, we changed ‘convincingly shows’ to ‘suggests’.

Round 2

Reviewer 3 Report (New Reviewer)

The authors have addressed most of my comments. There are only two issues left:

1. Although the authors responded that they calculated Shannon index for insect diversity at reclaimed vs control sites, and then removed it based on the reviewer’s recommendation in the first round, I still think it would be more appropriate here. From the authors’ response I see that the results were slightly different (higher diversity at reclamation sites only in the early season vs higher richness at reclamation sites in later season, non-significant difference in early season).  The consequences (interpretation) of this pattern may be different. However, I understand that there must have been a reason the reviewer in the first round recommended to remove the diversity measure and introduce richness instead (although I cannot think of any). I was invited later in the reviewing process; therefore, I will leave this to the editor’s assessment.

2. The authors have not addressed my comment on discussing dispersal abilities of the shared insect families, and their discussion on this topic is only superficial (lines 299–300). Although I still think discussing this could be very interesting and beneficial, I do not know how extensive or detailed should “communications” papers be. Therefore, I will leave it to the editor’s decision.

This manuscript is a resubmission of an earlier submission. The following is a list of the peer review reports and author responses from that submission.

Round 1

Reviewer 1 Report

This study focusses on influence of the restoration practices of disturbed areas (in particular cultivating the seeds of commercially available seed mix to the gas fields) in insect abundance and (family level) richness. While the topic of the study is very relevant and actual and has a strong applying side, there are at least three major issues that are not (fully) explained in the text, but that could strongly impact the results. I do not see the possibility to evaluate the manuscript appropriately before these issues are fully addressed.

1)    The Methods section lacks entirely the important explanation of how and according to what criteria the reference sites were chosen. Information presented in Introduction (L103-107) even suggests that these reference areas were chosen knowingly because they are known to have a little proportion of flowers and therefore little resources for pollinators! But this cannot be the criteria for such a comparison as the results seemed to be decided before the study and one can always found a reference ecosystem (e.g. woodland, bushland, marsh etc) nearby that conveniently produces the results one desires.

2)    The other major issue I see is related to the seed mix used. Although it is stated that that the mix consisted of native plants, it is not clear if these plants (except for Cleome serrulata) are common in the particular area and are adapted to the local environmental conditions or not. The usage of plants that are not common in the region/area may introduce at least two potential problems: 1) with a few years they may be outcompeted by the plant species that are better adapted to the local environment and therefore using the plants not adapted to local conditions may not be sustainable; 2) although the flowers of these species may be suitable resource for the adult insects as the nectar-feeders are mostly generalists, they may not be suitable for the larvae who are frequently specialists on particular plant species. Thereby these non-local plant communities may actually act as traps for pollinators by attracting the adults, but being unsuitable for the larvae.

3)    Do you determined only the adults or also the larvae from the insects collected? Please justify your approach in Methods.

Minor remarks

L25: „insect abundance and diversity”; also L47, L71 – diversity is a wider term that often also includes the abundance of organisms. “Richness” is a more specific and more suitable here.

L38: „A total of 237 insects…”, L39: “…858 insects…” – individuals? Pease specify!

L92: Please start the new paragraph with the main objective of the study

L115: “Well pads in this study averaged 5.3 acres…”  Please also indicate a range of sizes for the reclaimed areas and the average size and range of sizes of the reference areas.

L129-L131: Why this particular mix was chosen? Are the plants in the mix common in the study area?

L165-166: “Family-level diversity is typically similar to species-level diversity in terrestrial ecosystems” You mean richness (i.e. the number of species or families) here?

L175: “…sites were separated sites…” – unclear, second “site” redundant?

L190-193 scientific names should be in italic

L198-201 – marginally non-significant result (partly) treated as significant!

L207-213 – please specify that you mean the number of individuals here;

L223-225 – the wording here is confusing; do you mean the abundance of individuals within the families?

Reviewer 2 Report

In this study, the authors compared insect abundance and family richness between six natural gas well pads in reclaimed sites where a seed mixture was established and six well pads in adjacent reference areas. While the topic is of great interest, and the study could be a starting point for further research, several issues impede me from recommending this manuscript for publication.

1)      The hypothesis of this study concerns the diversity of insect communities (L106). Yet, insects were identified only to the family-level, which is a serious limitation for a community ecology study. Balmford et al. 1996 is cited to justify that higher taxonomy can be a surrogate for species richness, but this article focused on angiosperms, mammals and birds not on insects.

2)      Only two days of sampling were carried out, and it is unrealistic to expect that the insect community was thoroughly sampled. The sample size is just too small.

3)      In paragraph 2.1, Study Area, there is no information about the adjacent reference sites compared to the well pads. Moreover, how far were the sampling points from each other? Can they be considered independent? A map might clarify these issues, but currently, the study cannot be replicated.

4)      Two sampling days were carried out in June and July. However, there is no temporal replication. Moreover, these results are not comparable (L107-108) because different sites were sampled (Fig. 1 is misleading). Instead, 1- and 3-year-old pads should have been sampled in both June and July. This issue was recognised for the plant sampling (L173-174) but not for the insect sampling.

5)      The study is too short to assess whether the insect community has reestablished or not. One possible reason is mentioned in the introduction: the relative abundance of flowering species in the seed mixtures changes over the year. Therefore, with only one sampling season, it is impossible to claim that this seed mixture is effective (maybe insects were attracted by Cleome serrulata only in the first year after the seed mixture establishment, when this plant species was abundant).

6)      Assuming that the sampling sites were independent, the statistical analysis should consider the treatment (reclaimed vs. control area) as random factor and the replicate sites as a random factor. Moreover, more appropriate distribution should be used for the insect analysis (Poisson or Negative Binomial in case of overdispersion). Before analysing insect communities, rarefaction curves should be used to assess whether the sampling effort was enough, which is highly unlikely here.

Minor suggestions:

L25-26 Results from this study should be in the past tense also in the Simple Summary.

The final part of the introduction could be shortened, as L96-103 would be better placed in M&M.

L129-131 This was mentioned already.

L150 full stop missing.

L157 is it relevant where was the vehicle parked?

L164 identified (not capitalised)

L173, 186 here mentioning Figure 1, which is results, is inappropriate.

L207-208 “Insect abundance was significantly higher on reclaimed areas than adjacent reference ones (Figure 2).”

L208-209 Measure of dispersion (e.g., standard deviation) should be included when presenting the means.

L222 Chalcididae

L229-234 How can you compare early and late seasons if the sampling sites were not the same?

Paragraph 3.1 Species names should be in italics.

Suppl. Material “Early Season Insects” – the last row of data is missing. Moreover, the two files could and should be combined and reorganised.

Reviewer 3 Report

Thank you for submission of this well-written, original work.

Introduction: With the exception of the objective statement ("The objective of this study was to compare insect communities on natural gas well pads in the Jonah Infill natural gas field (Sublette County, Wyoming, USA) undergoing restoration activities at two points in the growing season to determine if insect communities were similar at different parts of the growing 95 season while different floral communities were in bloom.") segments of the final paragraph of the introduction would be better placed in the methodology. In fact there is significant and superfluous repetition of information therein, in the methods, including sampling dates. Elimination of this excess text from the introduction leaves room for a further explanation of the following points initially mentioned in the introduction (~ 250 words ea.), which would give the reader a stronger foundation for context of this work:

 1)     While traditional focus of insects in ecological restoration projects has been towards pollinators in agriculture systems, more recently they are becoming more common in non-crop studies [8, 11, 13-14].

 2)     In Wyoming, oil and gas operating companies are required to reclaim surface disturbances after extraction activities take place while complying with multiple regulatory criteria [15, 16].

 3)     The first study to examine how insects respond to ecosystem restoration efforts associated with natural gas development in a sagebrush-steppe ecosystem suggests these efforts increase insect abundance and diversity

 4)    The mass flowering hypothesis

The aforementioned 'objective' statement should then immediately precede the methods section.

Discussion:

Paragraph 1 (L.243), insert "that" between "shown" and "yarrow"